# Associations of Health Literacy with Blood Pressure and Dietary Salt Intake among Adults: A Systematic Review

**DOI:** 10.3390/nu13124534

**Published:** 2021-12-17

**Authors:** Darwish Mohd Isa, Suzana Shahar, Feng J. He, Hazreen Abdul Majid

**Affiliations:** 1Centre for Population Health, Department of Social and Preventive Medicine, Faculty of Medicine, University of Malaya, Kuala Lumpur 50603, Malaysia; darwish@um.edu.my; 2Faculty of Health Sciences, National University of Malaysia, Kuala Lumpur 50300, Malaysia; suzana.shahar@ukm.edu.my; 3Wolfson Institute of Preventive Medicine, Barts and The London School of Medicine and Dentistry, Queen Mary University of London, London E1 2AD, UK; f.he@qmul.ac.uk; 4Faculty of Public Health, Universitas Airlangga, Jawa Timur 60115, Indonesia

**Keywords:** health literacy, blood pressure, hypertension, salt intake, adult, systematic review

## Abstract

Health literacy has been recognized as a significant social determinant of health, defined as the ability to access, understand, appraise, and apply health-related information across healthcare, disease prevention, and health promotion. This systematic review aims to understand the relationship between health literacy, blood pressure, and dietary salt intake. A web-based search of PubMed, Web of Science, CINAHL, ProQuest, Scopus, Cochrane Library, and Prospero was performed using specified search/MESH terms and keywords. Two reviewers independently performed the data extraction and analysis, cross-checked, reviewed, and resolved any discrepancies by the third reviewer. Twenty out of twenty-two studies met the inclusion criteria and were rated as good quality papers and used in the final analysis. Higher health literacy had shown to have better blood pressure or hypertension knowledge. However, the relationship between health literacy with dietary salt intake has shown mixed and inconsistent findings. Studies looking into the main four domains of health literacy are still limited. More research exploring the links between health literacy, blood pressure, and dietary salt intake in the community is warranted. Using appropriate and consistent health literacy tools to evaluate the effectiveness of salt reduction as health promotion programs is required.

## 1. Introduction

Cardiovascular diseases (CVDs) remain one of the worldwide leading causes of death, leading to the deaths of 17.9 million people annually [1]. Heart attacks and strokes are responsible for 80% of CVD deaths, with one-third of these deaths happening in those younger than 70 years old highlighted in the World Health Organisation (WHO) report [2]. However, the relative risk from CVD deaths can be significantly reduced by managing blood pressure. [3]. Looking at the current trend, it is estimated that approximately 1.5 billion people will have hypertension by 2025 [4]. Similarly, CVDs remain the major cause of mortality in Malaysia, according to the Departments of Statistics Malaysia, 2020 [5]. Three in ten or 6.4 million people in Malaysia were reported to have hypertension [6]. One of the global goals for reducing non-communicable diseases (NCDs) is to reduce hypertension prevalence to 25% by 2025, against a 2010 baseline.

Excessive salt intake is the main contributing factor towards hypertension, with strokes accounting for 62% and coronary heart disease for 49% [7]. According to the WHO, the most cost-effective public health strategy to lower NCDs includes salt reduction [8]. In addition, it is one of the top three priorities for addressing the worldwide NCD crisis. Blood pressure can be lowered by reducing salt consumption among hypertensive and normotensive people, and it works in conjunction with antihypertensive medications [9]. Evidence has shown that the risk of CVD and death can be minimized by lowering salt consumption.

Between 2003 and 2011, population salt intake had reduced by 15% with the implementation of the United Kingdom’s salt reduction policy, where 85 product categories were set to lower salt content via independent monitoring. It appeared that there was a decrease in CVD mortality [10]. Following the UK’s lead, 96 countries currently have some salt reduction strategy in place [11]. Despite WHO recommendations of dietary salt intake below 5 g per day, salt intake remains high globally [12]. In Malaysia, outside foods are the well-known major contributor to high salt intake among the population. However, there is barely any movement or efforts made to address the issue [13].

Looking at the current trends, health literacy research is scarce. Health literacy is defined as the ability to access, understand, appraise, and apply health information in everyday life to make judgments and decisions about healthcare, disease prevention, and health promotion to maintain or improve one’s quality of life over time [14]. The concept of health literacy has risen in importance over the last two decades as a result of its numerous benefits to individuals, public health, and the healthcare system as a whole [14,15,16,17,18] and was even considered as one of the crucial determinants of health. Many studies have clearly shown the adverse health outcomes for health illiterate people, such as health issues, inefficient use of healthcare, increasing barriers to care, and early mortality [19]. In addition, an earlier systematic review has also found that limited health literacy hypertensive patients tend to have poor hypertension knowledge [20]. However, it is not clear about the general population level of health literacy regarding hypertension and salt intake itself. Available evidence suggested inconsistent and mixed findings between health literacy with blood pressure control and dietary salt intake among adults.

One of the most cost-effective strategies to fight NCD is to improve health literacy [19,21]. However, research on health literacy towards blood pressure control, salt intake, and other topics has shown mixed and contradictory results. Hence, the focus of this research is to learn more about the relationships between health literacy and dietary salt intake and blood pressure in adults.

## 2. Materials and Methods

### 2.1. Eligibility Criteria

A systematic review was conducted using the Preferred Reporting Items for Systematic Reviews and Meta-Analyses reporting criteria (PRISMA 2020) and was designed according to PICOS (Participant, Intervention, Comparison, Outcomes, and Study Design) criteria (Table 1). The review protocol (PROSPERO Registration number. CRD42021243596) has been registered with the International Prospective Register of Systematic Reviews.

A systematic literature search was conducted to identify studies that reported on the relationships between health literacy and blood pressure and dietary salt intake in people over the age of 18 years. It focused on the association between health literacy with dietary salt and blood pressure, using quantitatively validated instruments for health literacy and dietary salt and blood pressure. For intervention studies, only explicit interventions aiming at lowering blood pressure and dietary salt were considered. Studies that did not use any health literacy tools did not report any outcome of interest, were not published in English or Mandarin, were rated as poor or fair quality, or involved patients with specific conditions were all excluded.

### 2.2. Information Sources

A web-based search of PubMed, Web of Science, CINAHL, ProQuest, Scopus, Cochrane Library, and Prospero was performed. Databases search was supplemented with grey literature, internet searches (e.g., Google Scholar), reference lists of studies included in the systematic review, and manual search. The last search was run on 1 March 2021.

### 2.3. Search Strategy

The search was limited to English and Mandarin literature, published from January 2010 to March 2021, and will be re-run prior to the final analysis. Keywords and MESH terms, “health literacy” or “literacy” OR “numeracy” with “salt” or “salty” or “sodium” or “hypertension” or “blood pressure” were utilized and combined with the search using Boolean terms such as “AND” and “OR” (Table 2).

### 2.4. Selection Process

Articles were chosen in three stages: selection based on titles, followed by abstract consideration, and assessing the full text. Bibliographic information such as author, publication year, title, and journal, study design, setting, country, inclusion and exclusion criteria, subject recruitment, age, gender, and the study’s duration and dates were all acquired. Any mandarin language papers were translated by one of the reviewers.

### 2.5. Data Collection Process

The data gathered were then exported to Microsoft Excel from Mendeley, a reference manager software, and full texts. Two reviewers independently performed the data extraction and analysis, cross-checked, and reviewed and resolved any discrepancies by the third reviewer.

### 2.6. Data Items

To meet the objective of the review (i.e., understanding the associations between health literacy with blood pressure and dietary salt intake), tools utilized for assessing subjects’ health literacy, the appropriateness of blood pressure and salt intake measurement, and the outcomes of the associations were included. Other variables such as sociodemographic characteristics, knowledge, attitude, practice (KAP), and nutritional status were also extracted. The results of the measurements and the statistical methods used to assess the associations between them were then recorded along with the related conclusion and recommendations. All outcomes compatible with the outcome domain were sought in each study. The original authors of the studies were contacted when more information about the study outcomes or other details were needed.

### 2.7. Study Risk of Bias Assessment

Observational studies were assessed using the Newcastle–Ottawa quality assessment scale adapted for cross-sectional and cohort studies, measuring the selection, comparability, and outcomes. For interventions studies, random sequence generation, allocation concealment, selective reporting, other bias, blinding of participants and personnel, blinding of outcome assessment, and missing outcome data were all assessed using Cochrane’s collaboration tool. Quality scores were rated as “Good”, “Fair” or “Poor” quality. Each study’s risk of bias was independently assessed by two reviewers, and any disagreements were discussed with the third author. The bias assessment was reported in the tabulation.

### 2.8. Outcome Measures

Binary and continuous outcomes were gathered, and effective measures such as mean, mean difference, and odds ratios of the outcomes were used to synthesize and present results. The review also included other calculations/statistics, such as quartiles.

### 2.9. Synthesis Methods

The results were performed in tabulation and visual display of methods. A narrative synthesis of findings detailing the association between health literacy with blood pressure and dietary salt intake was performed.

## 3. Results

### 3.1. Study Selection

After eliminating duplicates from the databases, 1960 articles were identified. A total of 39 full texts were analyzed after the title and abstract screening. Twenty-two studies out of 39 met the requirements for inclusion, while 17 were excluded as they did not fit the criteria, did not report any outcome of interest, did not use any health literacy instrument, and involved patients with specific diseases other than hypertension. After quality assessment, two were removed as they were poor or fair quality papers [22,23], and the remaining 20 studies were included for synthesis. The PRISMA flow diagram of the literature search is shown in Figure 1.

### 3.2. Study Characteristics

There were 18 cross-sectional studies, one cohort study, and one RCT among the 20 papers included. The studies were conducted in United States (*N* = 7), Brazil (*N* = 2), Turkey (*N* = 2), Switzerland (*N* = 1), China (*N* = 1), Japan (*N* = 2), Iran (*N* = 2), Singapore (*N* = 1), Thailand (*N* = 1), and Cambodia (*N* = 1). All of the studies illustrate the relationship between health literacy with either blood pressure control or salt intake. Seventeen papers measured the associations between health literacy and blood pressure control, while three papers measured salt intake. Participants in studies include adults aged 18 years and above and the elderly, mostly with hypertensive conditions (*N* = 17). All studies used validated health literacy instruments though they varied considerably. Health literacy was most commonly measured by using Rapid Estimate of Adult Literacy in Medicine (REALM), a short version of the Test of Functional Health Literacy in Adults (STOFHLA), Newest Vital Sign (NVS), and Brief Health Literacy Screening (BHLS). The majority of studies or tools used to assess health literacy did not include all four main domains of health literacy; the ability to access, understand, appraise and apply health information. Out of 20 studies included, most studies assessed one’s ability to understand (*n* = 15), followed by appraise (*N* = 9), and apply (*N* = 9) health information, however very limited is known about the ability to access (*N* = 6) and the overall four domains of health literacy (*N* = 5). Most studies only assessed three health literacy domains at most.

For the outcome measurement, most studies used blood pressure measurement to assess blood pressure control. However, methods of salt measurement were diverse, including 24 h urine sodium (gold standard) and a twelve-item scale that assessed nutrition practices based on a low salt, DASH diet, and HL-SR (Health Literacy on Sodium Restriction). Table 3 lists the general characteristics of each study as well as the findings.

### 3.3. Quality Assessment

Of the 22 papers included initially, 20 are graded as good, the other two as fair and poor quality, hence were excluded from synthesis [22,23]. Details summary of quality appraisal is illustrated in Table 4, Table 5 and Table 6. for cross-sectional studies, cohort studies, and randomized trials, respectively.

### 3.4. Health Literacy Status

Health literacy status varied. This could be due to the participants’ diverse backgrounds, as well as differences in the types of tools and reporting methods used. Most studies categorized health literacy into two or more groups; limited and adequate; limited, marginal, adequate; problematic, inadequate, sufficient, excellent. Four studies presented health literacy status as mean and standard deviation [25,30,31,41], while one study reported health literacy using quartile [29]. Most of the studies determined the associations between health literacy and outcomes using a categorical approach by defining hypertension and health literacy levels, and some use health literacy index [3] quartiles and mean scores against systolic and diastolic blood pressure [37,40], hypertension, or salt intake levels. One study compared the proportions of the individuals who controlled systolic and diastolic blood pressure to the target between groups after intervention [24].

### 3.5. Health Literacy and Outcomes

Table 7 summarises the associations between health literacy and the outcomes of 20 papers included. The outcomes were grouped into two categories: Blood Pressure and Salt. Blood pressure control (*N* = 17) was the most commonly done, followed by salt (*N* = 3). One study [3] discussed two salt outcomes: salt intake (24 h urine) and awareness. Two studies [30,36] out of 20 papers included hypertension or blood pressure knowledge as an outcome and were included in the table, making a total of 23 outcomes.

#### 3.5.1. Health Literacy and Blood Pressure (Blood Pressure Control and Knowledge)

There are two outcomes in the blood pressure category; blood pressure control and blood pressure/hypertension knowledge. The majority of studies looking at health literacy and blood pressure control found a significant positive association (*p* < 0.05; *N* = 12), while four studies reported non-significant associations [24,30,33,36]. Patients with high health literacy tend to have better adherence to medication [26,28,29,32,39] since they are more likely to ask medical questions and involved in patient decision-making [38]. However, a study by Willens et al. (2013) [37] reported conflicting findings. Low health literacy (focused on understanding and appraising health literacy domains) patients had better blood pressure control than those with high health literacy. Patients with inadequate health literacy had much more encounters and, at the very least, are more committed to regular health care, as evidenced by the increased frequency of clinic visits. In addition, a cohort study among primary care patients reported that elevated blood pressure was linked to limited health literacy only for those with undiagnosed hypertension, not among those diagnosed [35].

Two studies reported an association between health literacy and blood pressure/hypertension knowledge [36,41]. One study conducted among hypertensive patients in Singapore reported that higher health literacy resulted in better knowledge of hypertension but not blood pressure control [36], while Suon and Ruaisungnoen (2019) [41] reported that knowledge on hypertension and sodium restriction are strongly related to the level of health literacy on sodium restriction.

#### 3.5.2. Health Literacy and Salt Intake (Low Salt Diet Adherence, 24 h Urine, and Health Literacy Sodium on Sodium Restriction)

Three studies explored the association between health literacy and dietary salt intake. The majority of studies found non-significant associations (*p* > 0.05; *N* = 2). These include a study using gold standard salt intake measurement, 24 h urine collection [3], and a study by Hutchison et al. (2014) [42], which measured low salt diet adherence using a low salt oriented questionnaire and found a non-significant association with health literacy. However, one study [41] on sodium restriction health literacy found that literacy skills, as well as knowledge of hypertension and sodium restriction, and health professional communication, were all found to be strongly linked. However, there is little research that looks at the relationship between health literacy and salt intake.

### 3.6. Health Literacy Domains

Out of 20 studies, only five measured all four domains of health literacy according to Sørensen et al. (2012); one’s ability to access, understand, appraise and apply health information [3,24,25,28,32]. The majority of health literacy tools used in the review did not measure all four domains. The number of studies assessing each domain was as follows; the ability to access (*N* = 5), appraise (*N* = 9), apply (*N* = 9), and followed by understand (*N* = 15). A few tools assessed all four domains of health literacy, namely European Health Literacy Survey Questionnaire (HLS-EU-Q), European Health Literacy Survey Questionnaire 47-item (HLS-EU-Q47), Health Literacy for Iranian Adults (HELIA), and a tool adopted from Ishikawa et al. (2008) [16] assessing the functional, communicative, and critical skills of the participants.

## 4. Discussion

The causes of poor hypertension management are multifaceted, which include obesity [43], medication adherence [44], dietary and other lifestyle factors [45]. However, from this review, it appears that an individual’s health literacy is one of the contributing factors to uncontrolled hypertension. This systematic review identified 20 studies investigating the associations between health literacy with blood pressure control and salt intake. Health literate patients have better hypertension control, [25,26,27,28,29,31,32,35,39,40,46] and knowledge of hypertension [30,36] and sodium restriction [41,47]. However, there were limited and conflicting findings when it came to health literacy and salt intake, and making comparisons between studies difficult due to the different timing and assessment tools.

Although health literacy research has improved over the last decade, before 2012, there was no consensus on the definition or its conceptual dimensions, restricting measurement and comparison possibilities. In addition, there were 16 types of health literacy tools used in the included studies, with only five studies assessing the main four domains of health literacy [3,24,25,28,32] despite the studies being rated as good quality. Most studies assessed the ability to understand health information, with the least focusing on the domains of access, followed by appraise and apply. Access refers to the ability to seek, find, and obtain health information. For example, people with low health literacy tend to postpone or avoid necessary treatment or report difficulty finding a practitioner [46]. Furthermore, health education was less likely to be paid attention to by those with low literacy skills and, as a result, were less likely to manage their disease, according to a systematic study [17].

People may now access a wide range of medical information through smartphones. More significantly, regardless of the skills, anyone may now develop health information and make it available to others [47] which many citizens or patients may find overwhelming when they have to apply or make a well-informed decision about their medical treatment. They may even be at risk of making a judgment that is harmful to their health in extreme circumstances [48]. Furthermore, previous studies have shown that many individuals believe that sodium content on food labels is difficult to read, understand, and interpret or comprehend [41,49]. Most instruments used in health literacy research assessed functional health literacy skills, which refers to one’s ability to locate, read and understand health information [50]. There are only a few measured information appraisals, which are the ability to interpret, filter, judge, and analyze accessed health information, despite widespread agreement that it is a basic set of skills to acquire in today’s society [51].

These domains highlight the differences between health literacy and the well-known KAP studies that assessed the respondents’ knowledge, attitude, and practice. In comparison with health literacy, KAP does not assess the extent of health literacy which dives into the aspect of one’s ability to access, understand, appraise and apply health-related information in healthcare, disease prevention, and health promotion, which has been proven to be a critical health determinant and may have been overlooked until recent years. Therefore, limited studies using appropriate health literacy tools that include the leading four domains highlight the need for more research utilizing the domains in the future.

In terms of blood pressure control outcomes, higher health literacy patients have better control of hypertension, according to most research, and vice versa. However, these findings were inconsistent with a study done in three primary clinics in Nashville with approximately 23,483 hypertensive patients, where it appeared that patients with higher literacy tend to have higher systolic and diastolic blood pressure. Patients with lower health literacy, they claimed, were at least are more inclined to seek regular treatment than those with adequate health literacy, as seen by the higher frequency of clinic visits. However, because these factors may reduce the impact of health literacy, a sensitivity analysis was carried out to adjust the number of clinic visits, and the outcome was unaffected [37]. Surprisingly, a cohort study by McNaughton et al. (2014) [34] reported that low health literacy was only linked to those undiagnosed with hypertension, and most of the existing evidence concerns hypertensive patients. These findings raise concerns that low health literacy may contribute to the high prevalence of uncontrolled blood pressure and being linked to undetected hypertension. More studies looking at these aspects are required among the general healthy population.

There appears to be a lack of studies exploring the link between health literacy and salt intake currently. Therefore, more investigation is required to justify the associations. The current findings on the association are inconsistent [3,41,42]. Out of all salt-related outcomes from studies included, two were shown to be strongly associated with health literacy: salt awareness [3] and health literacy sodium restriction [41]; however, relationships between health literacy and 24 h urine [3] and low-salt diet adherence [42] were found to be insignificant. Furthermore, several tools used to quantify salt intake may not be robust enough to represent salt intake among the population. Salt intake estimates based on dietary surveys or recall are unreliable [52]. Considering approximately 90% of sodium intake is eliminated in the urine, 24 h urine collection is the “gold standard” method for determining salt intake. [53,54,55]. However, the procedure is difficult for researchers to implement and participants to follow, making 24 h urine collection impracticable in many cases [53]. In this systematic review, there is only one study where the salt intake was measured using 24 h urine collection among the workplace community in Switzerland [3]. There was no significant link between health literacy index and salt intake. However, the sample size was constrained because of the organizations’ low response rate, which may have influenced the result. This finding is consistent with Hutchison et al. (2014) [42], where a low salt diet-oriented questionnaire was utilized.

Interestingly, individuals with adequate health literacy had a 6% higher chance of adhering to the low salt diet (OR = 1.06, 95% CI: 0.36–3.10) than those with limited health literacy, but the results remained non-significant. Otherwise, literacy skills were found significantly associated with health literacy sodium restriction. However, because the survey was more health literacy oriented, the tool utilized may not be powerful enough to reflect one’s salt intake. Based on current evidence, it has shown promising findings that health literacy could potentially be associated with understanding salt intake but not actual sodium intake that requires a robust tool to evaluate it appropriately, such as from urinary excretion or habitual dietary intake. It is important to highlight that the significant factor in the raised blood pressure is a high salt intake, which is accountable for CVD such as strokes and heart attacks [56]. The strong association between hypertension and dietary sodium has been widely recognized in several studies. [57,58,59].

This systematic review also discussed the relationship between health literacy and hypertension or hypertension knowledge. Health illiterate individuals are likely to have poor hypertension knowledge. Hypertension knowledge and health literacy were found to be significantly linked according to two studies in the review [36,41]. Ko et al. (2013) [36] reported that hypertension knowledge scores were significantly associated with health literacy level. Similarly, a cross-sectional study among hypertensive patients in Cambodia reported that when knowledge of hypertension and sodium restriction improved by one point, the health literacy score increased by 0.266 [41]. This finding demonstrated a link between these two parameters. Previous studies have found that persons who have enough knowledge about their health and salt intake are more inclined to minimize their salt intake for health reasons [60,61]. Patients with higher health literacy, according to many researchers, have more knowledge [62,63]; this may be due to the fact that health literacy skills express one’s ability to access and understand health information [64,65]. Furthermore, to understand printed education materials, patients would need to have higher health literacy. Even if patients with low health literacy could follow the instructions with the picture, they may have a problem communicating with healthcare professionals [66].

The collective evaluation revealed some noteworthy gaps in this area of research, including more research, which is needed to better understand the relationship between health literacy and salt intake, scarcity of evidence among the general population, and appropriate health literacy tools that assess all the crucial domains of health literacy.

### Strengths and Limitations

To the best of our knowledge, our study is the first of its kind to have systematically reviewed and discussed the lack of utilization towards the main domains of health literacy. Furthermore, existing evidence and gaps were identified, and aside from quality assessment, to reduce the risk of bias, the review included grey literature.

Our review also has some limitations. The scope is limited by the available evidence in the literature. Most studies included were cross-sectional, which limits the ability to identify the causal relationship. Cohort or intervention studies should be taken into consideration in the future to produce more valuable findings. In addition, the majority of studies were done among hypertensive individuals. Studies on the general population’s health literacy are still limited. There were only a few studies conducted investigating the associations between health literacy and salt intake. The variation of tools and methods used to interpret the results and outcomes were some of the limitations during the review process. These variations rendered comparison across studies difficult.

## 5. Conclusions

It appears that the majority of hypertensive individuals with higher health literacy tend to have better blood pressure control. However, the evidence of health literacy on salt intake itself is still scarce. Future research is needed to assess the applicability of health literacy with salt intake and blood pressure in other resource-limited settings.

## Figures and Tables

**Figure 1 nutrients-13-04534-f001:**
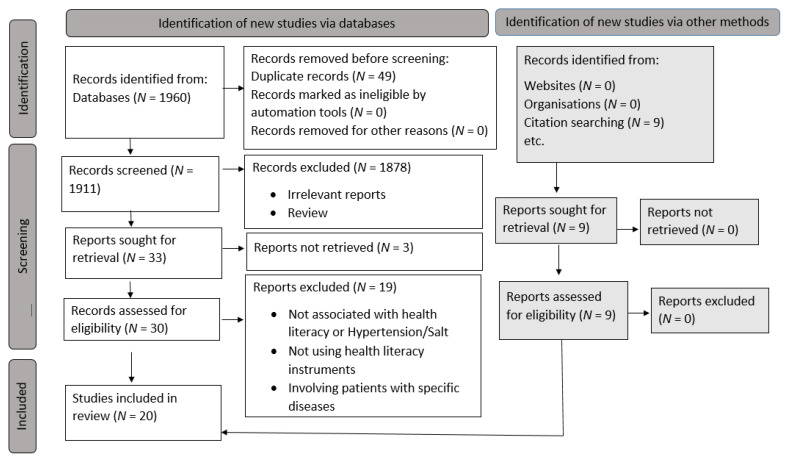
Flow diagram for study selection.

**Table 1 nutrients-13-04534-t001:** The PICOS criteria used to construct the systematic review.

Criteria	Description
Participants	Adults aged ≥18 years, including those with the hypertensive condition. Animal studies and individuals with other specific diseases were excluded
Intervention/Exposure	Health literacy
Comparison	High vs low health literacy in relation to the outcomes (blood pressure and dietary salt intake) either by mean scores, quartiles, or cut-off values such as limited, adequate, marginal, and excellent health literacy
Outcomes	Dietary salt intake and blood pressure using validated measurements and protocol
Study Design	Randomised controlled trial (RCT), non-RCT, cohort, and cross-sectional studies

**Table 2 nutrients-13-04534-t002:** Terms used for search strategy.

Concept 1	AND	Concept 2
Health Literacy OR Literacy OR Numeracy		Salt OR Salty OR Sodium Blood Pressure OR Hypertension

**Table 3 nutrients-13-04534-t003:** General characteristics of studies included in the final analysis.

**Author (Year)**	**Design**	**Instrument**	**Quality Score**	**Study Sample**	**Outcomes (Blood Pressure/Knowledge/Salt Intake)**
Delavar et al., 2020 [24]	RCT (January–March 2018)	Health Literacy for Iranian Adults (HELIA)	Good	Through block randomization, 118 older adults with uncontrolled hypertension were allocated to a control or intervention group at random. Age more than 60 years old.	Blood pressure: After Health Literacy-tailored intervention, blood pressure among the intervention group is improved; nevertheless, there was no evident difference between the groups (*p* > 0.05).
Gaffari-fam et al., 2020 [25]	Cross-sectional	HELIA	Good	210 hypertensive patients in Iran. Age more than 30 years (Mean age was 56.7 years)	Blood pressure: The health literacy dimensions contributed to a significant increase of 4.7% for the variance in high blood pressure.
Costa et al., 2019 [26]	Cross-sectional	The Short Assessment of Health Literacy for Portuguese-speaking Adults (18 items) SAHLPA-18	Good	392 hypertensive elderly patients. More than 60 years.	Blood pressure: Inadequate (high) blood pressure was linked to the following factor: inadequate functional health literacy.
Borges et al., 2019 [27]	Cross-sectional	Short Test of Functional Health Literacy in Adults (S-TOFHLA)	Good	357 adults from basic health units in Brazil. Aged between 18 to 39 years	Blood pressure: There was a statistically significant decrease in associations evaluated (*p* < 0.05) when it came to hypertension and participants’ health literacy level.
**Study, Year (References)**	**Design**	**Instrument**	**Quality Score**	**Study Sample**	**Outcomes (Blood Pressure/Knowledge/Salt Intake)**
Selcuk et al., 2018 [28]	Cross-sectional	European Health Literacy Survey Questionnaire (HLS-EU-Q)	Good	556 hypertensive patients in Turkey. Aged 18 years and above. Mean age was 55.74 ± 13.69 years (range 18–88)	Blood pressure: According to multivariate logistic regression analysis, health illiterate patients had higher uncontrolled blood pressure (OR: 2.06, 95% CI: 1.34–2.94).
# Halladay et al., 2017 [22]	Cohort	STOFHLA	Fair	493 patients with uncontrolled hypertension in rural primary care, US. The mean age was 57 (min = 20, max = 92) years.	Blood pressure: There were statistically significant reductions in mean Systolic Blood Pressure (SBP) in both the low and high health literacy groups (6.6 and 5.3 mmHg, respectively) after a year, however, there was no significant difference between the groups (Δ 1.3 mmHg, *p* = 0.067). The low and high health literacy groups both reported lower blood pressure in 2 years by 8.1 and 4.6 mm Hg, respectively, with no significant between-group difference (Δ 3.5 mm Hg, *p* = 0.25).
**Study, Year (References)**	**Design**	**Instrument**	**Quality Score**	**Study Sample**	**Outcomes (Blood Pressure/Knowledge/Salt Intake)**
# Shi et al., 2017 [23]	Cohort	Chinese health literacy scale for hypertension (CHLSH)	Poor	360 hypertensive patients in China. The age range of participants was 31–88 years.	Blood pressure: Low health literacy indicates high SBP. The rate of hypertension control increased as the CHLSH score increased (*p* < 0.001). The findings show that for three-quarters of the year, patients in the high literacy group have better SBP management than those in the low literacy group.
Hu et al., 2017 [29]	Cross-sectional	Health Literacy Scale for Hypertension	Good	596 hypertensive patients in China	Blood pressure: Blood pressure control was linked to total health literacy (*z* = 2.493, *p* = 0.013), ability to comprehend pictures (*z* = 3.187, *p* = 0.001), and accessing health-related information (*z* = 3.274, *p* = 0.001).
Yilmazel and Centikaya, 2017 [30]	Cross-sectional	Newest Vital Sign Scale and Blood Pressure Concept Test (adapted from REALM)	Good	500 volunteer teachers aged 35–49. The mean age of the study group was 42.91 ± 8.75 and in the hypertensive subjects, 48.35 ± 7.53.	Blood pressure: Health literacy was shown to be insignificant when it came to hypertension awareness and control.Knowledge: Those with hypertension who were aware of the disease had a higher health literacy level than those who were not (*p* > 0.05).
**Study, Year (References)**	**Design**	**Instrument**	**Quality Score**	**Study Sample**	**Outcomes (Blood Pressure/Knowledge/Salt Intake)**
Hall et al., 2016 [31]	Cross-sectional	SAHLSA (Short Assessment of Health Literacy for Spanish-Speaking Adults)	Good	45 Latino Migrant and Seasonal Farmworkers. Ages ranged from 29 to 60	Blood pressure: Higher levels of acculturation and health literacy were linked to improved blood pressure control (*p* = 0.01).
Wannasirikul et al., 2016 [32]	Cross-sectional	Adopted from Ishikawa et al. (2008)	Good	600 aged 60 to 70 years with a mean age of 65.3 years for hypertensive patients in Primary Health Care Centres in Thailand	Blood pressure: Blood pressure is strongly linked with health literacy (β = −0.14, *p* < 0.05).
Glashen, 2015 [33]	Cross-sectional	STOFHLA	Good	136 hypertensive Latino adults in the US aged 18 to 49 years	Blood pressure: Health literacy and hypertension association were not statistically significant (χ^2^ (1) = 0.811, *p* = 0.368).
McNaughton et al., 2014 [34]	Cross-sectional evaluation between 1 November 2010 and 30 April 2012	Brief Health Literacy Screen (BHLS)	Good	46,263 hospitalizations were available for analysis. Aged 18 years or older	Blood pressure: Low health literacy indicates extreme high blood pressure (aOR 1.08, 95% confidence CI 1.01, 1.16) and high blood pressure in people who had never been diagnosed with hypertension (OR 1.09, 95% CI 1.02, 1.16). Such associations were not found among patients with low health literacy and diagnosed hypertension.
McNaughton et al., 2014 [35]	Cross-sectional	The Rapid Estimate of Adult Literacy in Medicine (REALM)	Good	423 urban hypertensive patients with coronary disease in the US	Blood pressure: Limited health literacy indicates uncontrolled blood pressure (OR 1.75, 95% CI 1.06–2.87).
**Study, Year (References)**	**Design**	**Instrument**	**Quality Score**	**Study Sample**	**Outcomes (Blood Pressure/Knowledge/Salt Intake)**
Ko et al., 2013 [36]	Cross-sectional	STOFHLA Singapore	Good	306 hypertensive patients in the primary clinic in Singapore	Blood pressure: The degree of health literacy did not affect achieving the target blood pressure (*p* = 0.71). Knowledge: Higher health literacy level indicates higher hypertension knowledge scores (*p* < 0.001).
Willens et al., 2013 [37]	Cross-sectional	BHLS	Good	10644 hypertensive patients aged more than 18 years	Blood pressure: Health literate patients had a slightly lower odds of having their hypertension under control.
Aboumatar et al., 2013 [38]	Cross-sectional	REALM	Good	275 hypertensive patients in the US	Blood pressure: Patients with limited literacy reported poorer blood pressure management at the baseline.
Lenahan et al., 2013 [39]	Cross-sectional	TOFHLA	Good	215 hypertensive patients in the United States with an average age of 60 years old (SD = 8.0 years)	Blood pressure: Uncontrolled blood pressure (*p* = 0.03) and medication identification (*p* = 0.001) were both associated with health literacy.
Shibuya et al., 2011 [40]	Cross-sectional	Chinese Health Literacy (CHL)	Good	320 Middle-aged participants in an urban clinic, Japan (53 to 57 years) with an average age of 54.4 years old	Blood pressure: Limited health literacy and hypertension knowledge indicate poor health and raised blood pressure
**Study, Year (References)**	**Design**	**Instrument**	**Quality Score**	**Study Sample**	**Outcomes (Blood Pressure/Knowledge/ Salt Intake)**
Suon and Ruaisungnoen, 2019 [41]	Cross-sectional	Health Literacy Sodium Restriction (HL-SR)	Good	317 hypertensive patients in Cambodia. Age (21–72 years old) with average age of 54 years (SD = 8.95)	Salt Intake: Literacy skills (β = 0.125, *p* = 0.019), knowledge of hypertension and sodium restriction (β = 0.266, *p* < 0.001), and health professional communication (β = 0.359, *p* < 0.001) were reported to be strongly associated to Health Literacy-Sodium Restriction.
Luta et al., 2018 [3]	Cross-sectional	European Health Literacy Survey Questionnaire 47-item (HLS-EU-Q47)	Good	141 workplace population in Switzerland. Ages of 15 and 65	Salt Intake: The health literacy index and food literacy score did not have a significant relationship with salt intake (24 h urine), however, the awareness variable “salt content impacts food/menu choice” did.
Hutchison et al., 2014 [42]	Cross-sectional	Newest Vital Sign	Good	250 hypertensive patients in primary clinical care in the US. Age from 30 to 85 years (with an average age of 55 years).	Salt Intake: Adequate health literacy indicates a higher chance of adhering to the low salt plus diet (OR = 1.18, 95% CI: 0.50–2.79) than those with limited health literacy, but the results were not significant.

SD = standard deviation. # Studies that were excluded from synthesis.

**Table 4 nutrients-13-04534-t004:** Summary of Quality Assessment of the included studies using the Newcastle–Ottawa Quality Assessment Scale (adapted for cross-sectional studies).

Studies/Domains	Selection				Comparability	Outcome		
	Representativeness of the Sample	Sample Size	Non-Respondents:	Ascertainment of the Exposure (Risk Factor)	Comparability	Assessment of the Outcome	Statistical Test	Quality
Willens et al., 2013 [37]	*	*	*	**	**	**	*	Good (5 + 2 + 3)
McNaughton et al., 2014 [35]	*	*	*	**	**	*	*	Good (5 + 2 + 2)
Glashen, 2015 [33]								Good (5 + 2 + 2)
Gaffari-fam et al., 2020 [25]	*	*	*	**	**	*	*	Good (5 + 2 + 2)
Aboumatar et al., 2013 [38]	*	*	*	**	*	*	*	Good (5 + 1 + 2)
Suon and Ruaisungnoen, 2019 [41]	*	*	*	**	*	*	*	Good (5 + 1 + 2)
Yilmazel and Centikaya, 2017 [30]	*	*	*	**	*	*	*	Good (5 + 1 + 2)
Lenahan et al., 2013 [39]	*		*	**	**	*	*	Good (4 + 2 + 2)
Hutchison et al., 2014 [42]		*	*	**	**	*	*	Good (4 + 2 + 2)
Luta et al., 2018 [3]		*	*	**	**	*	*	Good (4 + 2 + 2)
Hall et al., 2016 [31]		*	*	**	**	*	*	Good (4 + 2 + 2)
Hu et al., 2017 [29]	*	*		**	**	*	*	Good (4 + 2 + 2)
Costa et al., 2019 [26]		*	*	**	**	*	*	Good (4 + 2 + 2)
Selcuk et al., 2018 [28]		*	*	**	*	*	*	Good (4 + 1 + 2)
Borges et al., 2019 [27]		*	*	**	*	*	*	Good (4 + 1 + 2)
Wannasirikul et al., 2016 [32]	*	*	*	*	*	*	*	Good (4 + 1 + 2)
Ko et al., 2013 [36]		*		**	**	*	*	Good (3 + 2 + 2)
Shibuya et al., 2011 [40]		*	*	*	**	*	*	Good (3 + 2 + 2)

Good quality: 3 or 4 stars in the selection domain, 1 or 2 two stars in the comparability domain, and 2 or 3 stars in the outcome/exposure domain. Fair quality: 2 stars in the selection domain, 1 or 2 stars in the comparability domain, and 2 or 3 stars in the outcome/exposure domain. Poor quality: 0 or 1 star in the selection domain, 0 stars in the comparability domain, and 0 or 1 star in the outcome/exposure domain.

**Table 5 nutrients-13-04534-t005:** Summary of quality assessment of the included studies using Newcastle-Ottawa Quality Assessment Scale (Cohort Studies).

Studies/Domains	Selection				Comparability	Outcome			
	Representativeness of the Exposed Cohort	Selection of the Non-Exposed Cohort	Ascertainment of Exposure	Demonstration that Outcome of Interest was not Present at the Start of the Study	Comparability	Assessment of Outcome	Was Follow-Up Long Enough for Outcomes to Occur	Adequacy of Follow-Up of Cohorts	Quality
McNaughton et al., 2014 [34]	*	*	*		**	*	*	*	Good (3 + 2 + 3)
# Halladay et al., 2017 [22]		*		*	**	*	*	*	Fair (2 + 2 + 3)
# Shi et al., 2017 [23]	*	*		*	**		*		Poor (3 + 2 + 1)

# Studies that were excluded from narrative synthesis; Good quality: 3 or 4 stars in selection domain AND 1 or 2 stars in comparability domain AND 2 or 3 stars in outcome/exposure domain. Fair quality: 2 stars in selection domain AND 1 or 2 stars in comparability domain AND 2 or 3 stars in outcome/exposure domain. Poor quality: 0 or 1 star in selection domain OR 0 stars in comparability OR 0 or stars in outcome/exposure domain.

**Table 6 nutrients-13-04534-t006:** Summary of quality assessment of the included studies using Cochrane’s collaboration tool for assessing the risk of bias in randomized trials.

	Random Sequence Generation	Allocation Concealment	Blinding of Participant and Personnel	Blinding of Outcome Assessment	Incomplete Outcome Data	Selective Reporting	Other Bias Due to Problems Not Covered Elsewhere	Quality
Delavar et al., 2020 [24]	Low risk	Low risk	Low risk	Unclear risk	Low risk	Low risk	Low risk	Good

Good quality: All criteria met (i.e., low for each domain) Fair quality: One criterion not met (i.e., high risk of bias for one domain) or two criteria unclear, and the assessment that this was unlikely to have biased the outcome, and there is no known important limitation that could invalidate the results Poor quality: One criterion not met (i.e., high risk of bias for one domain) or two criteria unclear, and the assessment that this was likely to have biased the outcome, and there are important limitations that could invalidate the results Poor quality: Two or more criteria listed as high or unclear risk of bias.

**Table 7 nutrients-13-04534-t007:** Health literacy and outcomes: summary of findings.

Category	Outcome	Design (Total Number of Studies by Design)	Positive Results (*p* < 0.05)	Negative Results (*p* < 0.05)	Non-Significant Results (*p* > 0.05)
Blood pressure	Blood pressure control	Cross-sectional (*N* = 15)	11	1	3
		Cohort (*N* = 1)	1		
		RCT (*N* = 1)			1
	Blood pressure/Hypertension knowledge	Cross-sectional (*N* = 2)	2		
Salt	Low salt diet adherence	Cross-sectional (*N* = 1)			1
	Salt awareness	Cross-sectional (*N* = 1)	1		
	24 h urine	Cross-sectional (*N* = 1)			1
	Health literacy sodium restriction	Cross-sectional (*N* = 1)	1		

## Data Availability

Data is contained within the article or supplementary materials.

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
