# Peer review of "Associations of Health Literacy with Blood Pressure and Dietary Salt Intake among Adults: A Systematic Review"

_nutrients, 2021, doi:10.3390/nu13124534_

Round 1

Reviewer 1 Report

This research article aims to understand the associations of health literacy with blood pressure and dietary salt intake among adults. The objective of a study is too simple. I wonder if the aim of this study meets the scope of Nutrient Journal. The number of investigated studies on salt intake is only three, which is not sufficient number of studies for authors to conclude the results. I wonder if authors attempted to search papers investigating the association between health literacy and other nutrient intake (or dietary patterns) related to blood pressure. In discussion, authors need to address risk factors of hypertension to clarify the only health literacy does affect risk of hypertension.

Author Response

Dear editors,

With this cover letter, we will submit the revised manuscript (Manuscript ID: nutrients-1432202) entitled, “Associations of health literacy with blood pressure and dietary salt intake among adults: A systematic review” for publication in Nutrients. We would like to thank referees for the careful and constructive reviews. Based on the comments from the referees, we have made changes of the manuscript, which are detailed below.

Reply to the evaluation by the first reviewer

  1. Comment 1: This research article aims to understand the associations of health literacy with blood pressure and dietary salt intake among adults. The objective of a study is too simple. I wonder if the aim of this study meets the scope of Nutrient Journal. The number of investigated studies on salt intake is only three, which is not sufficient number of studies for authors to conclude the results. I wonder if authors attempted to search papers investigating the association between health literacy and other nutrient intake (or dietary patterns) related to blood pressure. In discussion, authors need to address risk factors of hypertension to clarify the only health literacy does affect risk of hypertension.

Answer: Thanks for the comments. This study main focus was to assess the relationship between HL and salt intake and unfortunately not focussing on dietary patterns. Raised blood pressure is the leading cause of deaths and disability worldwide. High salt intake is the major cause of raised blood pressure. Our study aims to understand the relationship between health literacy and blood pressure, as well as salt intake. We feel such a study meets the scope of Nutrients. We also believe that the finding from our systematic review, i.e. there is a lack of studies on health literacy and salt intake, in itself, is important to highlight.

“It appears to be a lack of studies examining the relationship between health literacy and salt intake to date.”

“The causes of poor hypertension management are multifaceted which includes obesity [45], medication adherence [46], dietary and other lifestyle factors [47]. However, from this review, it appears that individual’s' health literacy is one of the contributing factors towards uncontrolled hypertension.”

“Based on current evidence, it has shown promising findings that health literacy could potentially be associated with understanding salt intake but not actual sodium intake that requires robust tool to evaluate it appropriately such as from urinary excretion or habitual dietary intake.”

We appreciate the comments from the reviewers. Thank you for reviewing our manuscript.

Reviewer 2 Report

Comments to the authors
This systematic review aims to understand the relationship between health literacy and, blood pressure and dietary salt intake. Blood pressure monitoring and dietary salt intake are important strategies for the prevention of cardiovascular diseases, and their relationship with health literacy is also an important point of view in preventive medicine. This systematic review is well structured and summarized, but several minor problems exist.

Minor points
1. Please verify the total number of papers for this review. There were 20 papers included in Figure 3.1, 22 papers in Table 3.2, and 23 papers in Table 3.5. The reviewer believes that there are repeated papers in Table 3.5, and if so, this information should be explained in the footnote.
Moreover, the authors described that the quality of 23 papers was assessed in 3.3 Quality Assessment section. The reviewer thinks that this information should reflect in the PRISMA flow diagram (Figure 3.1).
In 3.5 Health Literacy and Outcomes section, there were 22 papers.

2. Table 3.2: There are no explanations about the boldfaced expressions “CHLS,” “Z1/4,” and “P1/4.” Please add explanations in the footnote for Table 3.2. If boldface means excluding from synthesis, the reviewer ought to know why “Shi et al.” was indicated in boldface.

3. Table 3.2: In the study sample column for the reviewed paper by Ko et al. (2013), the authors described “306 patients.” Does it mean “306 patients with hypertension”? Please verify the characteristics of the study sample.

4. Table 3.2: The reviewer believes that it would be better if the authors could add the summary for the outcomes, such as blood pressure and salt intake, or both.

5. In 3.5.1 Health Literacy and Blood Pressure section, the authors used the word “interestingly” in the Results. The reviewer thinks that subjective expression should be avoided in the Results section.

6. Figure 3.1: “k” in the box at the top center should be “N,” please check and revise.

7. Page 1, lines 34–36: Please add the reference here.

8. Page 20, line 297: The authors described “one study” here. Please add the relevant citation.

9. Page 23, lines 394–395: The authors described “health literacy on sodium restriction (HL-SR)”, but this abbreviation is already used in line 203. 

10. Page 22, line 369: The authors referred the study by “McNaughton et al. (2014) [34]” as a cohort study, but reference number 34 might be a cross-sectional study according to Table 3.3.1. “[34]” should be “[33].” Please verify the references.

Author Response

Dear editors,

With this cover letter, we will submit the revised manuscript (Manuscript ID: nutrients-1432202) entitled, “Associations of health literacy with blood pressure and dietary salt intake among adults: A systematic review” for publication in Nutrients. We would like to thank referees for the careful and constructive reviews. Based on the comments from the referees, we have made changes of the manuscript, which are detailed below.

Reply to the evaluation by the second reviewer

  1. Comment 1: Please verify the total number of papers for this review. There were 20 papers included in Figure 3.1, 22 papers in Table 3.2, and 23 papers in Table 3.5. The reviewer believes that there are repeated papers in Table 3.5, and if so, this information should be explained in the footnote. Moreover, the authors described that the quality of 23 papers was assessed in 3.3 Quality Assessment section. The reviewer thinks that this information should reflect in the PRISMA flow diagram (Figure 3.1). In Table 3.5 Health Literacy and Outcomes section, there were 22 papers.

Answer: Thanks for highlighting this, for your information, 20 papers are included in review for synthesis (Figure 3.1), poor or fair quality papers were excluded as stated in the PRISMA figure (2 studies out of 22 were poor and fair quality papers, hence excluded)

In table 3.2, 22 papers were included for general summary (studies in bold are studies that were excluded for synthesis due to quality-added to footnote, but will be presented for general characteristics of papers) and also quality assessment (3.3.2)

Thus, after final calculation for Table 3.3, 22 papers were assessed

Table 3.5 summarises the associations between health literacy and the outcomes of 20 papers included. The outcomes were grouped into two categories: Blood Pressure and Salt. Blood pressure control (N = 17) was the most commonly done, followed by salt (N = 3). One study discussed 2 salt outcomes: salt intake (24-hour urine) and awareness. Two studies out of 20 papers include hypertension or blood pressure knowledge as an outcome, and were included in the table, making a total of 23 outcomes.

  1. Comment 2: Table 3.2: There are no explanations about the boldfaced expressions “CHLS,” “Z1/4,” and “P1/4.” Please add explanations in the footnote for Table 3.2. If boldface means excluding from synthesis, the reviewer ought to know why “Shi et al.” was indicated in boldface

Answer:

CHLS = Chinese health literacy scale for hypertension, added

Revised to Z= and P=, formatting issues

Footnote has been added, boldface means excluding from synthesis, Shi et al. and Halladay have been assessed for quality and were rated as “fair” and “poor”, therefore excluded from synthesis

Comment 3: Table 3.2: In the study sample column for the reviewed paper by Ko et al. (2013), the authors described “306 patients.” Does it mean “306 patients with hypertension”? Please verify the characteristics of the study sample.

Answer: Revised and verified, 306 patients were 306 hypertensive patients

  1. Comment 4: Table 3.2: The reviewer believes that it would be better if the authors could add the summary for the outcomes, such as blood pressure and salt intake, or both.

Answer: Revised, type of outcomes were stated before outcomes

  1. Comment 5: In 3.5.1 Health Literacy and Blood Pressure section, the authors used the word “interestingly” in the Results. The reviewer thinks that subjective expression should be avoided in the Results section.

Answer: Revised to “In addition, a cohort study…”

  1. Comment 6: Figure 3.1: “k” in the box at the top center should be “N,” please check and revise.

Answer: “K” has been revised to “N”

  1. Comment 7: Page 1, lines 34–36: Please add the reference here.

Answer: Citation has been included, (WHO, n.d)

  1. Comment 8: Page 20, line 297: The authors described “one study” here. Please add the relevant citation.

Answer: Citation added, (Ruaisungnoen, 2019)

  1. Comment 9: Page 23, lines 394–395: The authors described “health literacy on sodium restriction (HL-SR)”, but this abbreviation is already used in line 203. 

Answer: Revised into abbreviation “HL-SR”

  1. Comment 10: Page 22, line 369: The authors referred the study by “McNaughton et al. (2014) [34]” as a cohort study, but reference number 34 might be a cross-sectional study according to Table 3.3.1. “[34]” should be “[33].” Please verify the references.

Answer: Verified and yes, it was supposed to be [33], now [34] due to an additional references on the first page

We appreciate the comments from the reviewers. Thank you for reviewing our manuscript.
